# The Postnatal Offspring of Finasteride-Treated Male Rats Shows Hyperglycaemia, Elevated Hepatic Glycogen Storage and Altered GLUT2, IR, and AR Expression in the Liver

**DOI:** 10.3390/ijms22031242

**Published:** 2021-01-27

**Authors:** Paulina Kur, Agnieszka Kolasa-Wołosiuk, Marta Grabowska, Andrzej Kram, Maciej Tarnowski, Irena Baranowska-Bosiacka, Sylwia Rzeszotek, Małgorzata Piasecka, Barbara Wiszniewska

**Affiliations:** 1Department of Histology and Embryology, Pomeranian Medical University (PMU), Powstańców Wlkp. 72 Avene, 70-111 Szczecin, Poland; paulina.kur@pum.edu.pl (P.K.); sylwia.rzeszotek@pum.edu.pl (S.R.); barbara.wiszniewska@pum.edu.pl (B.W.); 2Department of Histology and Developmental Biology, Pomeranian Medical University, Żołnierska 48 Street, 71-210 Szczecin, Poland; martag@pum.edu.pl (M.G.); mpiasecka@ipartner.com.pl (M.P.); 3Department of Pathology, West Pomeranian Oncology Center, Strzałowska 22 Street, 71-730 Szczecin, Poland; akram@onkologia.szczecin.pl; 4Department of Physiology, Pomeranian Medical University, Powstańców Wlkp. 72 Avenue, 70-111 Szczecin, Poland; maciejt@pum.edu.pl; 5Department of Biochemistry and Medical Chemistry, Pomeranian Medical University, Powstańców Wlkp. 72 Avenue, 70-111 Szczecin, Poland; ika@pum.edu.pl

**Keywords:** glycogen storage, GLUT-2, AR, IR, finasteride, DHT deficiency, serum androgens and glucose concentration, zones of hepatic lobules

## Abstract

Background: A growing body of data indicates that the physiology of the liver is sex-hormone dependent, with some types of liver failure occurring more frequently in males, and some in females. In males, in physiological conditions, testosterone acts via androgen receptors (AR) to increase insulin receptor (IR) expression and glycogen synthesis, and to decrease glucose uptake controlled by liver-specific glucose transporter 2 (GLUT-2). Our previous study indicated that this mechanism may be impaired by finasteride, a popular drug used in urology and dermatology, inhibiting 5α-reductase 2, which converts testosterone (T) into dihydrotestosterone (DHT). Our research has also shown that the offspring of rats exposed to finasteride have an altered T–DHT ratio and show changes in their testes and epididymides. Therefore, the goal of this study was to assess whether the administration of finasteride had an trans-generational effect on (i) GLUT-2 dependent accumulation of glycogen in the liver, (ii) IR and AR expression in the hepatocytes of male rat offspring, (iii) a relation between serum T and DHT levels and the expression of GLUT2, IR, and AR mRNAs, (iv) a serum glucose level and it correlation with GLUT-2 mRNA. Methods: The study was conducted on the liver (an androgen-dependent organ) from 7, 14, 21, 28, and 90-day old Wistar male rats (F1:Fin) born by females fertilized by finasteride-treated rats. The control group was the offspring (F1:Control) of untreated Wistar parents. In the histological sections of liver the Periodic Acid Schiff (PAS) staining (to visualize glycogen) and IHC (to detect GLUT-2, IR, and AR) were performed. The liver homogenates were used in qRT-PCR to assess GLUT2, IR, and AR mRNA expression. The percentage of PAS-positive glycogen areas were correlated with the immunoexpression of GLUT-2, serum levels of T and DHT were correlated with GLUT-2, IR, and AR transcript levels, and serum glucose concentration was correlated with the age of animals and with the GLUT-2 mRNA by Spearman’s rank correlation coefficients. Results: In each age group of F1:Fin rats, the accumulation of glycogen was elevated but did not correlate with changes in GLUT-2 expression. The levels of GLUT-2, IR, and AR transcripts and their immunoreactivity statistically significantly decreased in F1:Fin animals. In F1:Fin rats the serum levels of T and DHT negatively correlated with androgen receptor mRNA. The animals from F1:Fin group have statistically elevated level of glucose. Additionally, in adult F1:Fin rats, steatosis was observed in the liver (see Appendix A). Conclusions: It seems that treating male adult rats with finasteride causes changes in the carbohydrate metabolism in the liver of their offspring. This can lead to improper hepatic energy homeostasis or even hyperglycaemia, insulin resistance, as well as some symptoms of metabolic syndrome and liver steatosis.

## 1. Introduction

The liver is one of the organs responsible for glucose metabolism. It is involved in the production of glucose and its storage as glycogen—a branched polymer of glucose produced mostly in the liver, in skeletal muscles and, to a smaller degree, in the brain. Glucose can be converted into glycogen and vice versa other via synthesis or degradation involving various steps in the glycogen metabolism pathways [1]. Glucose is transported in various ways including by glucose transporters (GLUT). In humans there are 14 isoforms (GLUT-1–GLUT-14) with diverse binding affinities and different expression profiles, that’s why are responsible for differences in glucose uptake by various human tissues [2]. Glucose absorption and release is dependent on the current needs of the body, and in hepatocytes, takes place mainly through the activation of GLUT-2 [3].

The liver is heterogenic in terms of the hepatocyte glycogen content, depending on the location of hepatocytes in the zones of the hepatic lobules. The zones differ in oxygenation, metabolic activity, and the phenotype of cells [4,5]. In zone 1 (also known as the periportal zone), glycogen forms large discrete aggregates, while glycogen in zone 3 (also known as the perivenous zone) is dispersed homogenously within the cytoplasm [6]. Glucose uptake and glycolysis occur intensively in zone 3, and the intensity of these processes decreases towards zone 1. In contrast, glucose delivery and gluconeogenesis (the generation of glucose from certain non-carbohydrate carbon substrates) are most intense in zone 1 and decrease in intensity towards zone 3 [6].

Glycogen accumulation in the liver can be diet-dependent [7,8,9] and age-dependent [9], e.g., younger mice store slightly more glycogen than older ones [10]. Moreover, an in vitro study showed that the primary culture of hepatocytes derived from old rats (around 24 months) exhibited a higher potential for glucose production when compared with the hepatocytes derived from younger rats (around 4-months old) [11]. Many data link increased hepatic glucose production to type 2 diabetes (T2D) [4].

Carbohydrate metabolism is also affected by levels of sex hormones [12,13,14]. Low testosterone (T) levels lead to reduced glucose tolerance and insulin activity (insulin resistance), particularly in elderly men [15,16]—effects which are correlated with type 2 diabetes. Testosterone therapy has been shown to significantly improve glucose tolerance and T2D in men [17]. Moreover, independent of obesity and metabolic syndromes (MetS) in men, testosterone deficiency is also associated with impaired fasting glucose and glucose intolerance [18]. Generally, testosterone is considered a key factor in gender related metabolic syndrome [19]. The lack of T in castrated rats causes symptoms similar to T2D or MetS, e.g., increased hepatic glucose synthesis (hyperglycaemia) as a consequence of inhibited insulin secretion, Akt phosphorylation, glucose uptake, glycogen synthase activity, GLUT-2 over-expression and glycogen phosphorylase activity in the liver. Supplementing castrated rats with testosterone or T with estradiol (E2) normalized the level of GLUT-2 mRNA and protein expression in the liver, whereas supplementation with E2 alone had no effect. Since normalization the testosterone level improved GLUT-2 expression, it can be presumed that T directly influences the *GLUT-2* transcription and translation [20]. Kelly et al. [21] also confirmed that exogenous T can stimulate synthesis of glycogen in both castrated and non-castrated rats. In humans, a high level of testosterone is related to a low risk of diabetes in men but a high risk in women [22,23].

The irreversible reduction in T into dihydrotestosterone (DHT) is carried out by 5α-reductase (EC1.3.99.5) [24]. 5α-reductase isozymes 1 and 2 are well studied [25]. It was reported that 5α-reductase-knockout mice (5αR1^−/−^, but not 5αR2^−/−^) on an American lifestyle-induced obesity syndrome (ALIOS) diet had a decreased hepatic mRNA expression in the genes involved with insulin signaling and developed hepatic steatosis [26]. The imbalance between T and DHT could be also caused by the inhibition of 5α-reductase type 2 by medicaments such as finasteride used by men suffering from benign prostatic hyperplasia (BPH), prostate cancer, and androgenic alopecia (AGA) [27,28]. Men with premature balding represent a risk group for the development of impaired glucose tolerance or T2D [29]. It was also shown that in male Zucker rats with genetic obesity (castrated and non-castrated), the finasteride treatment results in hyperinsulinemia [30]. Male 5αR1-knockout mice on a high-fat diet (HFD) showed a higher average weight gain and hyperinsulinemia comparing to wild animals. This may suggest a lack of activity in 5α-reductase and induces insulin resistance [30].

However, androgen concentration is not the only factor which influences glucose homeostasis. A key role is also played by the androgen receptor (AR)—a target for T, DHT, and other androgens. Many studies indicate that AR deletion contributes to the development of late visceral obesity with leptin resistance, insulin resistance and increased lipogenesis in adipose tissue and the liver [14]. A lack of androgen receptors in males promotes insulin resistance that could promote T2D development. An experiment carried out on hepatic AR-knockout mice fed a HFD showed that male H-AR^−/y^ (not female H-AR^−/−^) were overweight and were characterized by reduced sensitivity to insulin as a result of an increased expression of the protein-tyrosine phosphatase 1B (PTP1B), negative regulator of the insulin signaling pathway. So, the hepatic androgen receptor (as a positive factor), could also play an important role in avoiding insulin resistance development [31].

In our previous study on the same animal model, we showed that the offspring (F1:Fin) of rats exposed to finasteride had altered levels of serum androgens (T, DHT) and adverse changes in the morphology and physiology of testes and epididymides [32,33]. Given this trans-generational effect of finasteride on the male reproductive system and the aforementioned information on the role of androgens and AR in glucose metabolism, the aim of this study was to assess whether the androgen (T, DHT) imbalance in the F1:Fin generation of rats from males receiving finasteride can affect the accumulation of glycogen in the liver (androgen-dependent organ), serum glucose concentration and the hepatic mRNAs and proteins, GLUT2, IR, and AR expression.

## 2. Results

### 2.1. Percentage of PAS-Positive Glycogen Area in the Liver

#### 2.1.1. Glycogen Detection in Histological Section of the Liver Stained with PAS

Figure 1 shows PAS-positive granules indicating glycogen accumulation within hepatocytes. In adult rats (90 PND) of the F1:Fin group, we observed steatosis of the liver (Figure 1, insert on J). No signs of steatosis were visible in control rats of the same age (Figure 1, insert on I). Generally steatosis was characterized by fat accumulation, which is most prominent in the centrilobular zone.

#### 2.1.2. Percentage of PAS-Positive Glycogen Area in Whole Hepatic Lobules, without Dividing the Liver Parenchyma into Zones

The percentage of PAS-positive glycogen area in the liver (without dividing the lobules into zones) from F1:Control and F1:Fin groups of rats was differentiated by individual age groups. This value was statistically significantly higher in the F1:Fin rats in all age groups (Figure 2).

#### 2.1.3. Percentage of PAS-Positive Glycogen Area in the Zones of the Hepatic Lobules

The same relationship was observed in each of the individual liver zones (zones 1, 2, and 3). The percentage of PAS-positive glycogen area in each zone (periportal, intermediate, perivenous) of hepatic lobules in the livers of F1:Fin rats was statistically significantly higher in comparison to control animals (Figure 3). Moreover in the livers of F1:Fin as well as F1:Control, zone 1 contains less glycogen than zone 2. Zone 3 has the highest content of glycogen.

### 2.2. Relationship between Immunoexpression of GLUT-2 and Percentage of PAS-Positive Glycogen Area

Figure 4 shows the comparison between F1:Control and F1:Fin (in all age groups) of GLUT-2 immunoexpression according the IHC reaction intensity (weak, moderate, strong) within hepatocytes.

Statistically significant changes in the weak GLUT-2 expression was observed in 7 PND and 90 PND (higher and lower in F1:Fin vs. F1:Control, respectively). Taking into account the moderate expression of GLUT-2, the differences between F1:Fin vs. F1:Control rats were noticeable in 7 PND, 14 PND, 21 PND (higher), and 90 PND (lower). In the case of cells with a strong of GLUT-2 expression, the relationship changed. In the groups of 7 PND, 21 PND and 28 PND F1:Fin rats, the percentage of cells with a strong of GLUT-2 expression was significantly decreased vs. F1:Control, as opposed to 90 PND where F1:Fin animals showed a much greater pattern of expression than adult control rats (Figure 4).

Figure 5 shows liver with weak, moderate and strong GLUT-2 immunoexpression.

The Spearman’s rank correlation coefficient showed no correlation between percentage of PAS-positive glycogen area and a strong level of GLUT-2 expression.

### 2.3. qRT-PCR and IHC Analysis of GLUT-2, IR, and AR Expression

#### 2.3.1. GLUT-2, IR, and AR mRNA Expression without Dividing into Age Groups

Regardless of the age of the animals, the transcripts of GLUT-2, IR, and AR levels in homogenates of F1:Fin rats’ livers were changed in comparison to F1:Control animals (Figure 6). A noticeably statistically significant decrease in mRNAs concerned GLUT-2 and IR.

#### 2.3.2. GLUT-2, IR, and AR Expression with Dividing into Age Groups

Taking the age of the animals into account, decreased levels of GLUT-2 transcript were noticed at postnatal day 7 and 28 (Figure 7A). An IHC reaction confirmed this tendency (Figure 7B).

The immature animals of F1:Fin group in 7 PND, 21 PND, and mature in 90 PND have statistically lower levels of IR mRNA (Figure 8A), which was consistent with the immunoexpression of this marker at the protein level (Figure 8B).

The expression of androgen receptors on mRNA level was statistically reduced only in adult F1:Fin rats (Figure 9A), and its decreased intensity in IHC reaction was not so evident (Figure 8B), however some hepatocytes in the livers of experimental rats show cytoplasmic immunoreactivity (brown coloration of hepatocytes, Figure 9B) with only a nuclear localization of the receptor in control animals.

### 2.4. Relationship between Serum Androgen Levels and GLUT-2, IR, and AR Transcripts

In our previous reports [32,33] conducting on the same experimental model we showed that, the male rats from F1:Fin group were characterized by a differing serum testosterone and dihydrotestosterone concentration in comparison to the F1:Control.

Within the F1:Control group of rats, the Spearman’s rank correlation coefficient did not show any statistically significant correlation between the serum T concentration and the levels of GLUT-2, IR, and AR transcripts. Any correlation with the above mentioned markers was not shown in the serum DHT level either (Table 1).

In the F1:Fin group of rats, a statistically significant strong negative correlation was only shown between T and AR mRNA, and similarly DHT was also negatively correlated with the AR transcript level (Table 1).

### 2.5. Serum Glucose Level and It Correlation with GLUT-2 mRNA

The male rats from F1:Fin group were characterized by a differing serum glucose level in comparison to the F1:Control (Table 2). The statistical differences for glucose were on development day 14, 21, and 28.

The Spearman’s rank correlation coefficient did not show any statistically significant correlation between the serum glucose concentration and the age of animals (F1:Control: *r_s_* = −0.096, *p* = 0.58; F1:Fin: *r_s_* = −0.18, *p* = 0.29). Similarly, there were no correlation between the serum glucose concentration and GLUT-2 mRNA (F1:Control: *r_s_* = 0.09, *p* = 0.59; F1:Fin: *r_s_* = 0.27, *p* = 0.12).

## 3. Discussion

The bi-directional passive transport of glucose across plasma membranes [34] via GLUT-2 is responsible for the glucose balance in the cell. GLUT-2 up-regulation plays a more important role in the export of glucose than in its import to the liver [2]. GLUT-2 expression is required for the physiological control of glucose-sensitive genes [34]. To ensure the proper expression of glucose-dependent genes in the liver, it is necessary to maintain a proper balance between intracellular and extracellular GLUT-2-dependent glucose concentrations [35]. Mutations in the GLUT-2 gene not only cause transient neonatal diabetes mellitus (which disappears after approximately 18 months) [36] but also may lead to very rare Fanconi–Bickel syndrome, a condition associated with hepatomegaly (increased liver mass) and glycogen accumulation, growth retardation, and renal Fanconi syndrome [37,38].

In this study, a statistically increased level of glycogen in the liver of F1:Fin rats did not correlate with an increased level of GLUT-2, but in addition to high glycogen accumulation, we also observed steatosis in the livers of the adult (90 PND) offspring from males receiving finasteride. Therefore, it can be hypothesized that, in addition to ‘the glycogen storage disease’, rats of F1:Fin generation could also suffer from fatty liver disease, as a metabolic syndrome [4]. Moreover, the elevated glycogen accumulation in the liver of F1:Fin rats could be the result of increased glucose concentration in their blood serum. As it is common knowledge, that glycogenesis (the synthesis of glycogen from glucose) is stimulated inter alia by insulin in response to elevated blood glucose levels [3]. However the exact cause of elevated glycogen content in the liver of F1:Fin rats—lack of degradation or increased synthesis—have to be evaluated in the future (see Appendix A
Figure A1 and Figure A2).

It is well known that androgen receptors are expressed in the livers of male and female humans and rodents, and its hepatic expression is sex-dependent: AR expression is 20 times higher in the liver of adult male rats than in females [39]. AR expression is noted as age dependent: lower before puberty and higher in postpubescent life [40], which is partially compatible with our results. Lower AR expression were found at 14 PND, 21 PND and 28 PND, than in 90 PND of F1:Control; in F1:Fin rats, the expression of AR gradually decreased during pre-puberty, and the expression of AR transcripts in this group of rats was negatively correlated with circulating T and DHT. Since many liver diseases are associated with steroid hormones like androgens and their receptors, the male F1:Fin rats may have developed some kind of liver disease.

Androgen-deficient rats, e.g., due to castration, have significantly increased mRNA and protein levels of GLUT-2 in the liver [20]. In our experiment, the F1:Fin rats had an increased level of serum T, decreased serum DHT levels, an elevated expression of GLUT-2, and much higher hepatic glycogen accumulation than the control F1 generation. These findings suggest that DHT, similarly to testosterone in Shen et al. [13], is crucial for glucose homeostasis by regulating hepatic glucose output, and that testosterone deprivation due to castration increases hepatic glucose output, induces hyperglycaemia, and develops symptoms seen in type 2 diabetes and metabolic syndrome. On the other hand, this greater content of glycogen in the livers of F1:Fin rats could have been a result, as was reported previously by Muthusamy et al. [20], of the up-regulation of GLUT-2 which plays a more important role in controlling glucose export out of the liver than into it. Moreover, the direction of glucose transport depends on glucose concentration and is regulated by hormonal factors [3]. Therefore, it is highly likely that the rats from the F1:Fin group in our study could develop a metabolic disease in the future.

Androgens achieve the genomic effect via activation of nuclear receptors, followed by binding to a specific DNA region, known as the androgen response element (ARE) motif, localized in its target gene [41]. Testosterone replacement therapy restores GLUT-2 expression in castrated rat livers suggesting that testosterone may have a direct effect on GLUT-2 transcription and translation [20]. In the promoter region of the GLUT-2 gene, the presence of androgen response elements (ARE) has not been identified yet—this is probably why we did not observe any correlation between androgen concentration and GLUT-2 mRNA expression. However, according to McEwan et al. [42], AR acts as an independent ligand-activated transcription factor or it may bind to some other coactivators [43,44] to increase GLUT-2 expression. Androgens could also achieve a biological effect via a receptor associated with the plasma membrane of the cell, a mechanism that has not been thoroughly researched [13]. Although in our study F1:Fin rats in each age group had increased concentrations of circulating testosterone, the level of AR transcript was decreased, and adult rats in this group (90 PND) also had cytoplasmic immunoexpression of AR in some hepatocytes. This is in line with the conclusion of Shen et al. [13], who stated that “the integration of nongenomic effects via membrane receptor signaling and genomic effects via nuclear receptor signaling of sex hormones is critical to produce the final sex hormone cellular outcomes”.

In our study, the expression of insulin receptors, both at the mRNA and protein level was reduced in the liver of rat offspring born from females fertilized by the finasteride-treated male rats. On the other hand, there were no correlations between IR transcript and the level of serum androgens. It is known that MetS is characterized by the inability of insulin to adequately suppress hepatic gluconeogenesis, leading to hyperglycaemia, hyperinsulinemia and eventually to T2D [4]. In our experiment, in addition to the insulin resistance (decrease expression of insulin receptor), we showed hyperglycaemia of F1:Fin rats, that has been not age- and GLUT-2 mRNA level-dependent, so this elevated serum glucose concentration could have been caused by the trans-generational influence of finasteride. This is why it is likely that F1:Fin rats developed symptoms similar to MetS or even T2D during their lifetime, the origin of which might have been manifested by an increased body weight of F1:Fin vs. F1:Control that we find in a previous study on the same experimental model [32]. However this mechanism will be in the future evaluated by us (see Appendix A). The observed overweight could have been also associated with the modulation of adipose tissue metabolism by androgens (mainly DHT), as indicated by microarrays analysis by Zhang et al. [45], and therefore could be the result of a changed T–DHT ratio [33].

In summary, disturbed glucose transport into the cell (change of GLUT2 expression at the mRNA and protein level) and its utilization (increased glycogen accumulation in hepatocytes) in the liver of offspring from male rats receiving finasteride could result from decreased insulin receptor expression, elevated serum glucose concentration, as well as from the dysfunction of androgen regulation and signalling (changed T–DHT ratio and decreased AR expression).

## 4. Experimental Section

### 4.1. Animals

The study was conducted on albino Wistar rats in accordance with Polish law and with the approval of the Local Ethics Committee for Scientific Experiments on Animals in Szczecin, Poland (Resolution no. 23/2010). Parent generation F0 produced male generation F1. Paternal rats in the F0:Fin group (*n* = 5) were treated with finasteride Proscar, MSD, Cramlington, UK) in daily doses of 5 mg/kg/bw, the same as in our previous studies [46,47,48] and experiments by other researchers [49,50]. The period of finasteride treatment before mating lasted 56 days (this changed the morphology and function of the testis) [46,47,48]. The mating period lasted a period of 5 months and not longer, to avoid the effect of ageing of the parent generation. Male rats (F0:Fin) received finasteride up until the end of the experiment (for 4–5 months). Once a week, the animals were weighed and the finasteride dose adjusted.

### 4.2. Generation of Filial Animals

The control group of offspring (F1:Control, *n* = 25) consist of male rats born from females fertilized by untreated control male rats. The experimental group of offspring (F1:Fin, *n* = 25) consists of male rats born from females fertilized by finasteride-treated male rats. The objective of the experiment was to sample the liver from both the treated and untreated offspring (F1) at 7, 14, 21, 28, and 90 postnatal days (PND) of life. Detailed information on the design of the animal treatment, mating, and collection of newborn offspring has been presented in our previous report [32]. After thiopental anesthesia (120 mg/kg bw, i.p., Biochemie GmbH, Vienna, Austria), the livers were divided into two parts, the first part was fixed in formalin and used for Periodic Acid-Schiff staining and immunohistochemical (IHC) reactions, the second tissue samples were frozen and used for qRT-PCR analyses.

### 4.3. Histological and Immunohistological Methods

The dissected livers were fixed in 10% formalin for at least 24 h and then washed with absolute ethanol (3 times over 3 h), absolute ethanol with xylene (1:1) (twice over 1 h) and xylene (3 times over 20 min). Then, after 3 h of saturation of the tissues in liquid paraffin, the samples were embedded in paraffin blocks. Using a microtome (Microm HM340E), 3–5 μm serial sections were taken and placed on polysine microscope slides (Thermo Scientific, UK; cat. no. J2800AMNZ). The sections of the livers were deparaffinized in xylene and rehydrated in decreasing concentrations of ethanol, and then used for PAS staining and IHC reaction.

PAS staining was made to demonstrate glycogen within the cell, because PAS reaction breaks 1,2-glycol linkages to form aldehydes, which are then revealed by Schiff’s reagent [51]. To perform this procedure was used commercial kit (Bio-Optica, cat. no. 04-130802). Positive staining was determined microscopically (Leica DM5000B, Wetzlar, Germany) by visual identification of magenta color granules within cytoplasm of hepatocytes.

In order to expose the epitopes to IHC procedure, the deparaffinized and rehydrated sections were boiled twice in Target Retrieval Solution (DacoCytomation, S2367, S2369) in a microwave oven (700 W twice for 5 min). Once cooled and washed with PBS, the endogenous peroxidase was blocked using a 3% solution of perhydrol in methanol, and then the slides were incubated over night at 4 °C with primary antibodies against: GLUT-2 (Invitrogen, PA5-77459, final dilution 1:250), AR (Santa Cruz Biotechnology, sc-7305; final dilution 1:50), and IR (Abcam, ab 60946, final dilution 1:20). Antibodies were diluted in Antibody Diluent with Background Reducing Components (Dako, S3022). To visualize the antigen-antibody complex, a Dako LSAB+System-HRP was used (DakoCytomation, K0679), based on the reaction of avidin-biotin-horseradish peroxidase with DAB as a chromogen, according to the staining procedure instructions included. Sections were washed in distilled H_2_O and counterstained with hematoxylin. For a negative control, specimens were processed in the absence of a primary antibodies. Positive staining was determined microscopically (Leica DM5000B, Wetzlar, Germany) by visual identification of brown pigmentation.

### 4.4. Quantitative Computer Image Analysis Histological Slides

PAS-stained and GLUT-2-immunostained slides were scanned at a magnification of 400× (resolution of 0.25 μm/pixel) using the ScanScope AT2 scanner (Leica Microsystems, Wetzlar, Germany). The obtained digital images of the slides were analysed using the ImageScope viewer (Version 11.2.0.780; Aperio Technologies, Inc., Vista, CA, USA).

For the automatic computer analysis of PAS-positive glycogen in rat livers, a positive pixel count algorithm (version 9.1; Aperio Technologies, Inc.) was used. Other parameters were set to achieve compliance with the visual evaluation of colour intensity, including the intensity thresholds for positive results. The areas of analyses were manually determined. Using the algorithm, the number of PAS-positive and PAS-negative pixels were counted. The total number of PAS-positive pixels was counted in 30 random fields for each of the three zones of the hepatic lobule in each studied group with an average area of 0.049 mm^2^ (6 fields per rat), Subsequently, the percentage of PAS-positive glycogen areas was calculated.

For the automatic computer analysis of GLUT-2 expression in the membrane of hepatocytes, a membrane v9 algorithm (version 9.1; Aperio Technologies, Inc.) was used. Similar to the PAS-positive glycogen analysis, other parameters were set to achieve compliance with the visual evaluation. The areas of analyses were also manually determined. Using the algorithm, the percentage of hepatocytes with weak, medium, and strong GLUT-2-positive immunostaining in the plasma membranes were counted. The percentage of cells with the GLUT-2 expression was counted in 30 random fields in each group with an average area of 0.217 mm^2^ (6 fields per rat).

### 4.5. Quantitative Real-Time Reverse Transcription PCR (qRT-PCR) Analysis

Quantitative analysis of mRNA expression of GLUT-2, IR, and AR were performed in a twostep reverse transcription PCR. Tissues were suspended in 600 μL of RLT buffer and homogenized for 4 min on ice. Next, the sample was digested with proteinase K for 15 min at 55 °C and isolated on spin columns according to the manufacturer’s protocol. All RNA was extracted from 50–100 mg tissue samples using an RNeasy Lipid Tissue Mini Kit (Qiagen, Hilden, Germany). The RNA was then treated with DNase I (Qiagen) to eliminate genomic DNA contamination. All isolated RNA was quantified by spectrophotometry using a NanoDrop ND-1000 spectrophotometer (Nano-Drop Technologies, USA) and the optical density 260–280 nm ratio was determined; the 260:280 ratios were 1.8–2.0. Next, cDNA was prepared from 1 μg of total cellular RNA in a 20 μL reaction volume, using a FirstStrand cDNA synthesis kit and oligo-dT primers (Fermentas, USA). Quantitative assessment of mRNA levels was performed by real-time RT-PCR using an ABI 7500 Fast instrument with a Power SYBR Green PCR Master Mix reagent. Real-time conditions were as follows: 95 °C (15 s), 40 cycles at 95 °C (15 s), and 60 °C (1 min). The specificity assessment was conducted by performing a melting curve analysis (60 to 95 °C in temperature ramp melting); only one PCR product was amplified under these conditions. Reaction mixtures contained 10 μL of 2× SYBR Green supermix, 3 μL of primers (0.4 μmol/L each), 2 μL of cDNA template and 3 μL of water Each sample was analyzed in two technical replicates, and mean *Ct* values were used for further analysis. The relative quantity of a target, normalized to the endogenous control Gapdh gene and relative to a calibrator, is expressed as 2–ΔΔ*Ct* (-fold difference), where *Ct* is the threshold cycle, Δ*Ct* = (*Ct* of target genes) − (*Ct* of endogenous control gene), and ΔΔ*Ct* = (Δ*Ct* of samples for target gene) − (Δ*Ct* of calibrator for the target gene). The following primer pairs were used: for GLUT-2 (F: TCA GAA GAC AAG ATC ACC GGA; R: GCT GGT GTG ACT ATG AGT GGG), for IR (F: ATG GGC TCC GGG AGA GGA T; R: CTT CGG GTC TGG TCT TGA ACA), and for AR (F: TCC AAG ACC TAT CGA GGA GCG; R: GTG GGC TTG AGG AGA GCC AT). The primer sequences used in the study were obtained according to the sequence information obtained from the NCBI database, and were synthesized by Oligo.pl (IBB PAN, Poland).

### 4.6. Hormone Assays

The procedure of measurement of T and DHT in blood serum is the same as in our previously published work [33]. A standard sandwich ELISA assay was performed on the serum using a rat specific T and DHT ImmunoAssay System kit (CUSABIO; CBS-E05100r and CBS-E07879r), according to the manufacturer’s instructions, and an Asys UVM 340 microplate reader (Asys Hitech Gmbh, Austria). The detection range for T was 0.13–25.6 ng/mL with a sensitivity of 0.06 ng/mL; for DHT, the detection range was 10–2000 pg/mL and the sensitivity 5 pg/mL.

### 4.7. Glucose Assesment in Blood

The levels of glucose in blood serum were measured in the Central Laboratory of The Independent Public Clinical Hospital No. 2 of Pomeranian Medical University in Szczecin (SPSK2, PUM). Glucose was measured by the UV test (700/340 nm), the reference enzymatic method with hexokinase (GLUC-3, Roche/Hitachi, cobas c 311, cobas c 501/502). The detection range was 2 to 750 mg/dL and the sensitivity 2 mg/dL.

### 4.8. Statistical Analysis

The results of the qRT-PCR analysis, serum glucose and androgens concentrations were analyzed using Statistica 6.1 software (StatSoft, Kraków, Poland). The arithmetical means and SDs (X ± SD) were calculated for each of the parameters. The normal distribution of the results for the individual variables was obtained using the Shapiro–Wilk test. As most of the distributions deviated from a normal distribution, nonparametric tests were used for further analysis. To assess the differences between the groups, the nonparametric Mann–Whitney *U*-test and Kruskal–Wallis test with Dunn’s multiple comparison test for post hoc analysis were used. A probability of *p* ≤ 0.05 was considered statistically significant. Additionally, the serum levels of T and DHT, presented in our previous study [33], were correlated with the levels of GLUT-2, IR, and AR transcripts by Spearman’s rank correlation coefficient (*r_s_*). The same test was used to correlate serum glucose concentration with age of animals and hepatic GLUT-2 mRNA.

## 5. Conclusions

Rats in the F1:Fin group have a higher accumulation of hepatic glycogen as a result of hyperglycaemia, a lower expression of GLUT-2, IR, and AR in the liver, and liver steatosis and higher body weight (documented it in a previous study on the same experimental model [32]).

It can be concluded that finasteride has trans-generational consequences and probably epigenetic side effects that could lead to some metabolic syndromes such as hyperglycaemia, insulin resistance, type 2 diabetes or a fatty liver in males. Moreover, finasteride should become the focus of research in the relatively new field of pharmacology—pharmacoepigenomics.

## Figures and Tables

**Figure 1 ijms-22-01242-f001:**
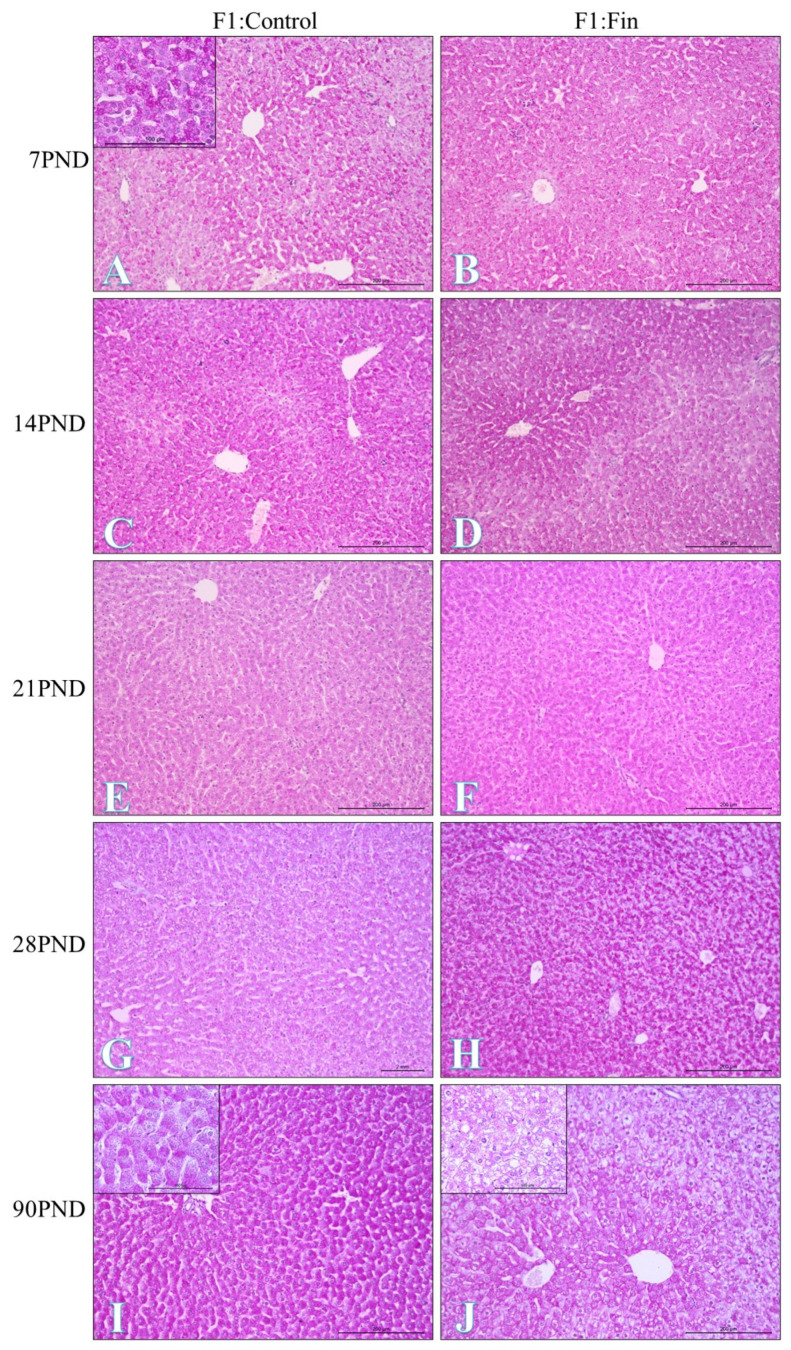
PAS-positive glycogen granules (magenta) within hepatocytes of rats offspring born from females fertilized by control (F0:Control) and finasteride-administrated (F0:Fin) male rats in their postnatal life (F1:Control/F1:Fin: 7 PND, 14 PND, 21 PND, 28 PND, 90 PND). Periodic Acid Schiff staining. Objective magnification: (**A**–**J**) ×20, scale bar 200 µm and insertion on (**A**, **I**, **J**) ×40, scale bar 100 µm.

**Figure 2 ijms-22-01242-f002:**
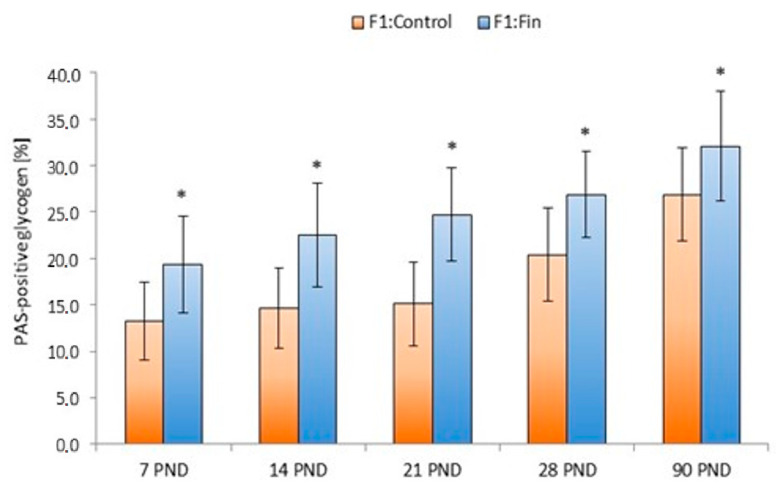
The percentage of PAS-positive glycogen areas in the livers of rats according to the days of development (7 PND, 14 PND, 21 PND, 28 PND, 90 PND) in the control (F1:Control) and in the F1 generation after finasteride administration (F1:Fin), * *p* < 0.001 vs. control in each group.

**Figure 3 ijms-22-01242-f003:**
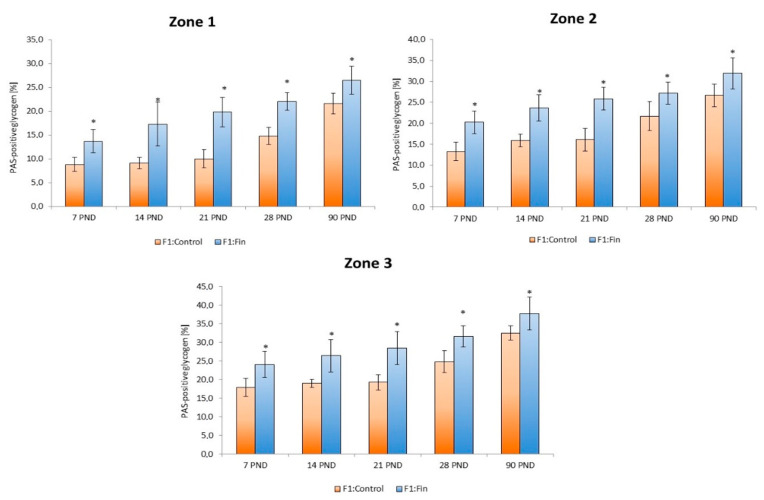
The percentage of PAS-positive glycogen areas in zones 1, 2, and 3 of the hepatic lobules of rats according to the days of development (7 PND, 14 PND, 21 PND, 28 PND, 90 PND) in the control (F1:Control) and in the F1 generation after finasteride administration (F1:Fin), * *p* < 0.001 vs. control in each group.

**Figure 4 ijms-22-01242-f004:**
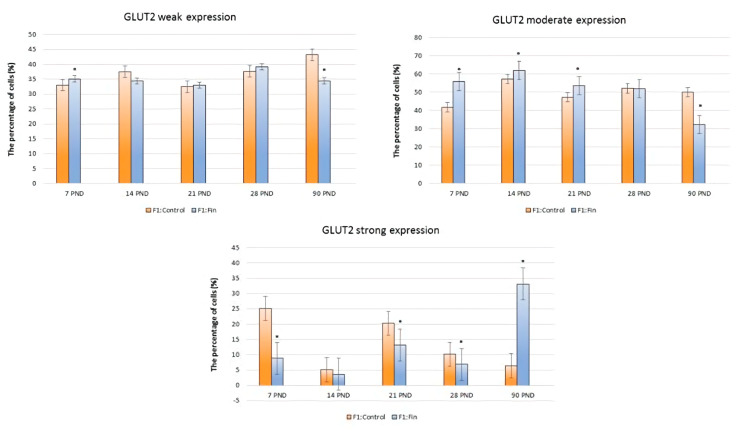
The percentage of GLUT-2-positive cells (divided into weak, moderate, and strong immunoexpression) in the livers of rats according to days of development (7 PND, 14 PND, 21 PND, 28 PND, 90 PND) in the control (F1:Control) and in the F1 generation after finasteride administration (F1:Fin), * *p* < 0.001 vs. control in each group.

**Figure 5 ijms-22-01242-f005:**
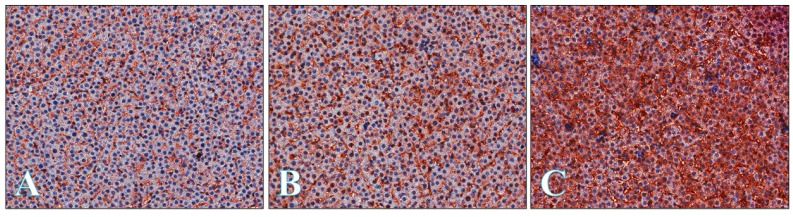
Representative microphotography showing GLUT-2 expression in the livers at the weak (**A**), moderate (**B**), and strong (**C**) level. Objective magnification: ×20.

**Figure 6 ijms-22-01242-f006:**
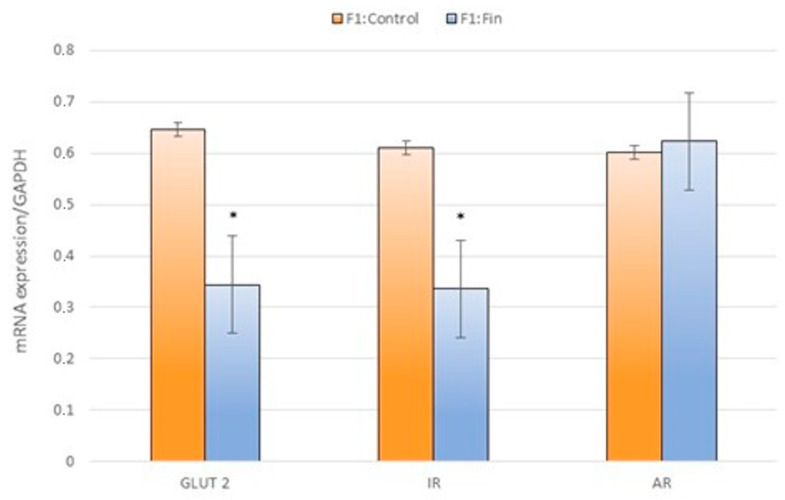
The glucose transporter 2 (GLUT-2), insulin receptor (IR) and androgen receptor (AR) mRNA levels (normalized to GAPDH) in the homogenates of hepatic tissue of the control offspring (F1:Control) and those born from females fertilized by finasteride-treated male rats (F1:Fin) without taking into account postnatal age. Values are expressed as arithmetic means ± SD; differences were evaluated using the Mann–Whitney *U*-test (*n* = 25 per each F1:control and F1:Fin,, *p* ≤ 0.05).

**Figure 7 ijms-22-01242-f007:**
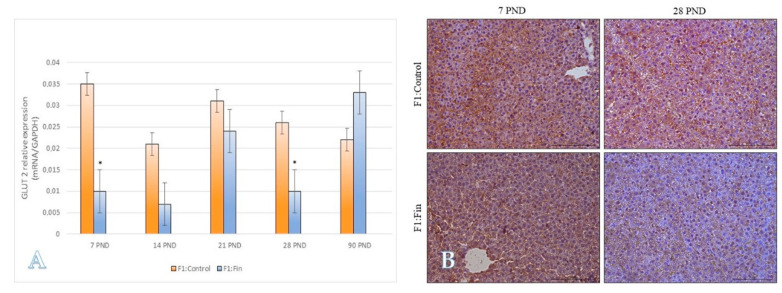
(**A**) The glucose transporter 2 mRNA levels (normalized to GAPDH) in the homogenates of hepatic tissue of control offspring (F1:Control) and those (F1:Fin) born from females fertilized by finasteride-treated male rats in postnatal days 7, 14, 21, 28, and 90. Values are expressed as arithmetic means ± SD; differences were evaluated using the Mann–Whitney *U*-test (*n* = 5 per each age group; *p* ≤ 0.05). (**B**) Representative microphotography comparing differences (according the qRT-PCR analysis of 7 PND and 28 PND) in immunoexpression of GLUT-2 within the liver of F1:Control vs. F1:Fin animals. IHC reaction. Objective magnification ×20, scare bar 100 µm.

**Figure 8 ijms-22-01242-f008:**
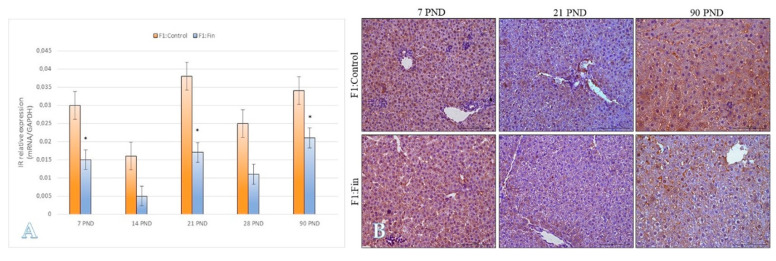
(**A**) The insulin receptor mRNA levels (normalized to GAPDH) in homogenates of hepatic tissue of control offspring (F1:Control) and those born (F1:Fin) from females fertilized by finasteride-treated male rats in postnatal days 7, 14, 21, 28, and 90. Values are expressed as arithmetic means ± SD; differences were evaluated using the Mann–Whitney *U*-test (*n* = 5 per each age group, *p* ≤ 0.05). (**B**) Representative microphotography comparing differences (according the qRT-PCR analysis of 7 PND, 21 PND, and 90 PND) in immunoexpression of IR within the liver of F1:Control vs. F1:Fin animals. IHC reaction. Objective magnification ×20, scare bar 100 µm.

**Figure 9 ijms-22-01242-f009:**
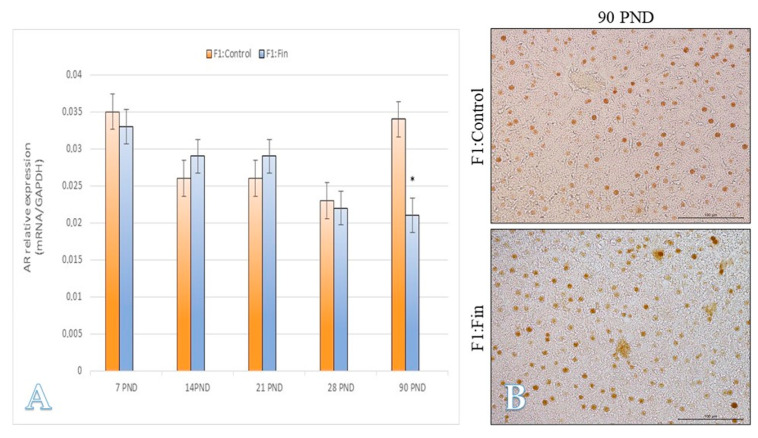
(**A**) The androgen receptor mRNA levels (normalized to GAPDH) in homogenates of hepatic tissue of control offspring (F1:Control) and those (F1:Fin) born from females fertilized by finasteride-treated male rats in postnatal day 7, 14, 21, 28, and 90. Values are expressed as arithmetic means ± SD; differences were evaluated using the Mann–Whitney *U*-test (*n* = 5 per each age group, *p* ≤ 0.05). (**B**) Representative microphotography comparing differences (according the qRT-PCR analysis of 7 PND and 28 PND) in immunoexpression of AR within the liver of F1:Control vs. F1:Fin animals. IHC reaction. Objective magnification ×20, scare bar 100 µm.

**Table 1 ijms-22-01242-t001:** Correlation between serum androgens (T, DHT) and mRNAs for glucose transporter 2, insulin receptor and androgen receptor in the liver homogenates of male control rats’ offspring (F1:Control) and finasteride-treated male rats’ offspring (F1:Fin).

	F1:Control		F1:Fin	
	T	DHT	T	DHT
**GLUT2**	*r_s_* = −0,000 *p* = 1.000	*r_s_* = −0.096 *p* = 0.655	*r_s_* = 0.009 *p* = 0.974	*r_s_* = −0.088 *p* = 0.689
**IR**	*r_s_* = 0.449 *p* = 0.081	*r_s_* = 0.012 *p* = 0.951	*r_s_* = −0.353 *p* = 0.235	*r_s_* = 0.067 *p* = 0.761
**AR**	*r_s_* = 0.143 *p* = 0.583	*r_s_* = 0.342 *p* = 0.102	*r_s_* = −0.762 *p* = 0.000375	*r_s_* = −0.562 *p* = 0.005253

Values are express as the Spearman’s rank coefficient (r_s_) with statistically significant probability *p* ≤ 0.05.

**Table 2 ijms-22-01242-t002:** Glucose concentrations in blood serum of rat offspring (F1:Control, F1:Fin, respectively) born from females fertilized by control (F0:Control) or finasteride-treated (F0:Fin) male rats.

Age	Glucose [mg/dL]	
PND	F1:Control	F1:Fin
14	118.0 ± 8.6	162 ± 29.3 ** vs. F1:Control
21	138.2 ± 72.0	177.9 ± 17.3 * vs. F1:Control
28	187.9 ± 19.1	200.6 ± 19.8 * vs. F1:Control
90	189.3 ± 13.3	190.0 ± 27.3

F1:Control, F1:Fin—rat offspring born from females fertilized by the control or finasteride-treated male rats, respectively; PND: postnatal day. Values are expressed as arithmetic means ± SD (n = 5 per age group) evaluated by the Mann–Whitney U-test; * *p* ≤ 0.05, ** *p* < 0.001.

## Data Availability

The data presented in this study are contained within this article and Appendix A.

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
