# Peer review of "The Postnatal Offspring of Finasteride-Treated Male Rats Shows Hyperglycaemia, Elevated Hepatic Glycogen Storage and Altered GLUT2, IR, and AR Expression in the Liver"

_ijms, 2021, doi:10.3390/ijms22031242_

Round 1
Reviewer 1 Report
The reviewed article is a resubmission of a previous manuscript (no 1000478), that has been already reviewed by myself. The Authors have followed my suggestion from the comments and made adequate correction/changes.
In the present manuscript the Authors added additional data to Results section (glucose level in serum), that allow to demonstrate a state of hyperglycemia in F1:Fin rats. It added a new value to results and discussion in the context carbohydrate metabolism. The Authors added also a set of microphotographs to section 2.2 (Figure 5) presenting the IHC for GLUT2 expression in F1:Control and F1:Fin rats in studied age. However more suitable for this section would be the representative microphotographs showing “weak”, “moderate” and “strong” immunoexpression of the GLUT2.
The Authors have additionally written a quite extensive Supplementary data (Appendix A) in which they present a set of new results connected with lipid metabolism in studied model. I must say that I feel a little bit confused with this section. I my opinion the data presented there are more suitable for writing a new article and not be supplementary data to the results presented in the manuscript. I will leave the decision as to the supplementary data to Editor.
Below there are few comments/suggestions concerning text editing:
Line 29 – change for “.. weather the administration of finasteride had an trans-generational effect on….
Line 31 and 32 – instead of “correlation” the better would be “relation”
Line 110 – should be “comparing to”
Line 380-381 – “The experimental group of offspring (F1:Control, n=25) consists of male rats born from….”
Author Response
Dear Reviewer,
Thank you very much for the effort in one more revision of our manuscript. We would also like to thank you for your valuable comments. We have corrected our manuscript and all changes You can see in manuscript (in blue color). Additionally, we decided according your comments to remove Fig. 5 that shows GLUT2 expression in the liver of animal according their day of life, instead we added new one, which show only “weak”, “moderate” and “strong” immunoexpression of GLUT2.
Best regards,
Agnieszka Kolasa-Wołosiuk
Reviewer 2 Report
Dear authors,
Thank you for this updated manuscript that you have provided.
- I believe that the data you added make a much stronger case for your manuscript. Furthermore, it shows that it is not only the glucose metabolism that is deregulated, but the treatment also deregulated lipid metabolism.
- While I still believe that the titles of the paragraphs of the Results section need to be more indicative of the result obtained in the given paragraph (to make it more clear to the reader), I leave this choice up to you and the editor.
- Please remove this last paragraph. ‘’Appendix B: Manuscript weaknesses: To more strongly confirm the trans-generational effect of finasteride on liver metabolism, in the future will be done more detailed analysis based on DNA microarrays, that will show the differences between F1:Control and F1:Fin in transcriptome (expression of genes related not only to carbohydrate metabolism but also lipid metabolism).’’ The reader does not need to have this information.
- I previously wrote ‘’The table 1 contains the same values as the Figure 1 and 2 in your already published paper ‘’Androgen levels and apoptosis in the testis during postnatal development of finasteridetreated male rat offspring’’, you need to cite the paper, not show the same results again.’’ To which you replied: We found the presentation with previously published results (references are in description of Table 1, [33]) as justified, since in this publication we correlated the levels of T and DHT with GLUT2, IR and AR mRNAs by Spearman’s rank correlation coefficient. We believe that leaving the table 1. in the publication will not be self-plagiarized.
I am confident that this data CAN NOT be added in your new manuscript as table 1, as THESE DATA WERE ALREADY PUBLISHED TWICE. You published these data in ‘’Androgen levels and apoptosis in the testis during postnatal development of finasteride treated male rat offspring’’ AND in ‘’Antioxidant enzyme expression of mRNA and protein in the epididymis of finasteride-treated male rat offspring during postnatal development’’. You can cite both of your papers and indicate which data you took to do the correlation (ex. Table III from [33]), but you cannot put the table itself. Again, I leave this to the judgement of the editor.
Author Response
Dear Reviewer,
Thank you for your effort to read and correct my manuscript. I would also like to thank you for your valuable comments.
I am agree with you, that added new data shows that not only glucose metabolism but also lipid metabolism was deregulated, and these two biochemical events will make my manuscript much more strong and professional.
According the Reviewer 1, we decided to remove Fig. 5 that shows GLUT2 expression in the liver of animal according their day of life, instead we added new one, which show only “weak”, “moderate” and “strong” immunoexpression of GLUT2 (please see in manuscript); and we hope it finds your approval.
We also removed the Appendix B: Manuscript weaknesses, as you recommended.
I edited ta little bit he titles of paragraph in Results section, but only a little bit, because they were corrected according recommendation of Reviewer 1 during the first round or revision (please see in manuscript, in blue color), and secondly, you left me and the Editor free choice.
Thank you one more time for your valuable revision.
Best regards,
Agnieszka Kolasa-Wołosiuk
This manuscript is a resubmission of an earlier submission. The following is a list of the peer review reports and author responses from that submission.
Round 1
Reviewer 1 Report
The reviewed article describes the effect of paternal exposure to 5-alpha reductase type 2 inhibitor (finasteride) on some aspects of glucose metabolism in liver of their male offspring. The analysis is performed in different stages of the offspring development on protein (histo- and immunohistochemical methds) and mRNA (qRT-PCR) levels. Authors concluded that paternal treatment with finseteride shows changes in hepatic glycogen storage and GLUT2, insulin and androgen receptors mRNAs expression which could lead to improper hepatic energy homeostasis insulin resistance as well as some symptoms of metabolic syndrome and liver steatosis. Moreover the observed changes are considered to results from decreased insulin receptor expression as well as dysfunction of androgen signaling (T/DHT imbalance and decreased AR expression) caused by paternal treatment.
The work is solid and represents a useful contribution to the literature concerning the sex-hormone dependent hepatic disease. The strength of the study is its transgenerational model. The methods and statistics are applied properly. The results are also of clinical importance, although a simple extrapolation of them on human beings shouldn’t be done. However, the manuscripts in parts needs some corrections and additional explanations to make it more readable (see comments).
Detailed comments:
Abstract:
- Line 30 – “finasteride treatment has a transgenerational effect…” not finasteride per se
- Line 32 - the 3rd point of the goal must be changed as if the finasteride treatment can’t affect “correlation” – correlation is a statistical relationship between two random variable
- Lines 34, 35 - the abbreviations for the study groups should be placed at the end of the sentences, just after explanations what they mean
- Line 35-37 –change for: “In the histological sections of liver the PAS staining (to visualize glycogen) and IHC (to detect GLUT2, IR and AR immunoexpression) were performed. The liver homogenates were used QRT-PCR to asses GLUT2, IR and AR mRNA expression.
Introduction:
- Line 57 – sentence need rewriting to be more understandable
- Line 82 – delete “(metabolic syndrome)” - the abbreviation is explained the line above
- Line 84 – GLUT over-expression….
- Line 87 – Since normalization of the testosterone level….
- Line 93 – change: “researched” for “studied”
- Line 100 – “TD2” instead “diabetes mellitus type 2”
- Line 103 - sentence need rewriting to be more understandable
Results:
- Line 124 – the title of the paragraph should be changed for more adequate. “Hepatic glycogen content” suggest rather a direct measurement of the level of glycogen in the liver tissue; in the article the semi-quantitative indirect analysis was performed on histological liver sections stained with PAS method utilizing of computer image analysis. The obtained results is the percentage of PAS-positive glycogen area of the liver and as such should be presented in the text
- Line 125 – the title should be changed– this paragraph presents a glycogen detection in histological section of the liver stained with PAS reaction.
- Line 127-128 –a few words concerning histological signs of observed steatosis should be added; what criteria were applied?
- Line 129 - the Figure 1 is too small and it is difficult to observed “granules indicating glycogen accumulation within hepatocytes”; it would valuable to add inserts with much larger maginification in each part of the figure
- Line 131 – the description of control and studied group should be more precise for eg. “PAS-positive glycogen granules (magenta) within hepatocytes of rat offspring born from females fertilized by control (F1:Control) and finasteride-administrated male rats (F1:Fin) in their postnatal life (7 PND, 14 PND, 21 PND, 28 PND, 90 PND)….”
- Line 134, 135, 137, 142, 144, 146 ect – see the comment above (point 12; line 124) - the text need rewriting to avoid expression like “glycogen content/level” etc.
- Line 139-141 – see comment above (point 16 – line 131)
- Line 153 – the title needs to be more precise – again the word “level” should be avoided; the method used is immunohistochemistry and again computer image analysis allows to differentiate the strength (intensity) of the signal (immunohistological reaction) as weak, moderate or strong in hepatocyestes (its plasma membrane). The result is the percentage of the cells with weak, moderate and strong immunoexpression of GLUT2. What does it mean “…against glycogen levels”?
- Line158-160 - see comment above (point 16 – line 131)
- Line 173 - GLUT2, IR and AR mRNA expression
- Line 183 - GLUT2, IR and AR mRNA expression in different age gropus
- Line 207 – “brown color coloration “ ?
- Line 2018 – “by a different…”
- Line 222-224– see comment above (point 16 – line 131)
- Line 237 and 238 – there is no need to repeat the rs and p values which are presented in the Table 2
Experimental section
- Line 326 and 328 - The control group of offspring (F1:Control, n=25) consists of male rats born from….
- Lines 336-338 – This part of the text should be added to the paragraph below,. Iassumed that the ptocess of fixation and preparing the liver sections were the same for PAS satininh as well immunohistochemical methods.
- Line 339 - the title of the paragraph should be changed for eg. “PAS staining and immunohistochemistry” or “Histological and immunohistochemical methods”
- Line 360 – 374 – in the paragraph there is no information about scanning of the slides with immunohistohemical reaction against GLUT2; It would be better to separate the description of PAS and GLUT image analysis to make the paragraph more understandable.
- Line 402 – the hormone measurement were made in serum or in plasma?
- Line 406-407 – delete “The T and DHT protein concentrations were normalized to total protein levels as measured by a BCA kit (Pierce, USA) using bovine albumin as a standard”
- Line 423 – in this article the body weight of the animals was not analysed; the mentioned data (line 303) concerns the previous studies, so it shouldn’t be mentioned in the conclusion in that way.
Author Response
Reviewer 1
Dear Reviewer,
Thank you very much for the work put in review of our manuscript. We would also like to thank you for your valuable comments and instructions. We have corrected our manuscript and all changes You can see below.
Best regards,
Agnieszka Kolasa-Wołosiuk
Detailed comments:
Abstract:
1. Line 30 – “finasteride treatment has a transgenerational effect…” not finasteride per se
Re: We changed into: …. to assess whether the administration of finasteride had a consequence in a transgenerational effect on……
2. Line 32 - the 3rd point of the goal must be changed as if the finasteride treatment can’t affect “correlation” – correlation is a statistical relationship between two random variable.
Re: It was not corrected, because the goal was to check whether the level of androgens correlates with the expression of GLUT1, IR, AR. But in the results it turned out that there is no such correlation, therefore we leave this sentence unchanged.
3. Lines 34, 35 - the abbreviations for the study groups should be placed at the end of the sentences, just after explanations what they mean.
Re: Abbreviations were replaced.
4. Line 35-37 –change for: “In the histological sections of liver the PAS staining (to visualize glycogen) and IHC (to detect GLUT2, IR and AR immunoexpression) were performed. The liver homogenates were used QRT-PCR to asses GLUT2, IR and AR mRNA expression”.
Re: This has been improved.
Introduction:
5. Line 57 – sentence need rewriting to be more understandable
Re: Sentence was rewritten.
6. Line 82 – delete “(metabolic syndrome)” - the abbreviation is explained the line above
Re: It has been deleted.
7. Line 84 – GLUT over-expression….
Re: It has been changed.
8. Line 87 – Since normalization of the testosterone level….
Re: It has been improved.
9. Line 93 – change: “researched” for “studied”
Re: It has been changed.
10. Line 100 – “TD2” instead “diabetes mellitus type 2”
Re: It has been changed.
11. Line 103 - sentence need rewriting to be more understandable
Re: It has been changed to: “Male 5αR1-knockout mice on a high-fat diet (HFD) showed a higher average weight gain and hyperinsulinemia compering to wild animals. This may suggest a lack of activity in 5α-reductase and induces insulin resistance”. The whole work was corrected by a native speaker.
Results:
12. Line 124 – the title of the paragraph should be changed for more adequate. “Hepatic glycogen content” suggest rather a direct measurement of the level of glycogen in the liver tissue; in the article the semi-quantitative indirect analysis was performed on histological liver sections stained with PAS method utilizing of computer image analysis. The obtained results is the percentage of PAS-positive glycogen area of the liver and as such should be presented in the text.
Re: According to suggestion the title has been changed in to: Percentage of PAS-positive glycogen area in the liver.
13. Line 125 – the title should be changed– this paragraph presents a glycogen detection in histological section of the liver stained with PAS reaction.
Re: According to suggestion the title was changed in to: Glycogen detection in histological section of the liver stained with PAS.
14. Line 127-128 –a few words concerning histological signs of observed steatosis should be added; what criteria were applied?
Re: The subject of research in this publication was the evaluation the carbohydrates metabolism, that’s why we only mentioned liver steatosis as an additional observation; more detailed data and criteria will be presented in the next work on lipid metabolism. So it hasn’t been changed. Generally steatosis is characterized by fat accumulation, which is most prominent in the centrilobular zone.
15. Line 129 - the Figure 1 is too small and it is difficult to observed “granules indicating glycogen accumulation within hepatocytes”; it would valuable to add inserts with much larger maginification in each part of the figure.
Re: The figure 1 was magnified.
16. Line 131 – the description of control and studied group should be more precise for eg. “PAS-positive glycogen granules (magenta) within hepatocytes of rat offspring born from females fertilized by control (F1:Control) and finasteride-administrated male rats (F1:Fin) in their postnatal life (7 PND, 14 PND, 21 PND, 28 PND, 90 PND)….”
Re: The description has been changed into: PAS-positive glycogen granules (magenta) within hepatocytes of rats offspring born from females fertilized by control (F0:Control) and finasteride-administrated (F0:Fin) male rats in their postnatal life (F1:Control/F1:Fin: 7 PND, 14 PND, 21 PND, 28 PND, 90 PND). Periodic Acid Schiff staining. Objective mag. A - J X20, scale bar 200 µm and insertion on A, I, J x40 scale bar 100 µm.
17. Line 134, 135, 137, 142, 144, 146 ect – see the comment above (point 12; line 124) - the text need rewriting to avoid expression like “glycogen content/level” etc.
Re: It has been improved.
18. Line 139-141 – see comment above (point 16 – line 131)
Re: It has been improved.
19. Line 153 – the title needs to be more precise – again the word “level” should be avoided; the method used is immunohistochemistry and again computer image analysis allows to differentiate the strength (intensity) of the signal (immunohistological reaction) as weak, moderate or strong in hepatocyestes (its plasma membrane). The result is the percentage of the cells with weak, moderate and strong immunoexpression of GLUT2. What does it mean “…against glycogen levels”?
Re: It has been changed.
20. Line158-160 - see comment above (point 16 – line 131).
Re: Generally we did not change the description of Figures because in our opinion they are informative. Additionally improved description of Fig 1. made it very long and apart from using F1Control/F1:Fin
we had to add F0:Control and F0:Fin.
21. Line 173 - GLUT2, IR and AR mRNA expression
Re: It has been changed.
22. Line 183 - GLUT2, IR and AR mRNA expression in different age groups
Re: We cannot add mRNA to the title of this subsection because the results from the IHC are here also discussed.
23. Line 207 – “brown color coloration “ ?
Re: Color has been removed.
24. Line 218 – “by a different…”
Re: Has been changed.
25. Line 222-224– see comment above (point 16 – line 131).
Re: Please check our comments at the point 20.
26. Line 237 and 238 – there is no need to repeat the rs and p values which are presented in the Table 2.
Re: We removed these duplicated information from text.
Experimental section
27. Line 326 and 328 - The control group of offspring (F1:Control, n=25) consists of male rats born from….
Re: It has been improved.
28. Lines 336-338 – This part of the text should be added to the paragraph below,. I assumed that the process of fixation and preparing the liver sections were the same for PAS staining as well immunohistochemical methods.
Re: It has been improved.
29. Line 339 - the title of the paragraph should be changed for eg. “PAS staining and immunohistochemistry” or “Histological and immunohistochemical methods”
Re: It has been changed.
30. Line 360 – 374 – in the paragraph there is no information about scanning of the slides with immunohistochemical reaction against GLUT2; It would be better to separate the description of PAS and GLUT image analysis to make the paragraph more understandable.
Re: It has been improved - the text is divided into paragraphs.
31. Line 402 – the hormone measurement were made in serum or in plasma?
Re: The hormone measurement were made in serum.
32. Line 406-407 – delete “The T and DHT protein concentrations were normalized to total protein levels as measured by a BCA kit (Pierce, USA) using bovine albumin as a standard”
Re: It has been deleted.
33. Line 423 – in this article the body weight of the animals was not analyzed; the mentioned data (line 303) concerns the previous studies, so it shouldn’t be mentioned in the conclusion in that way.
Re: We re-edit this conclusion and add the information about references to those data.
Reviewer 2 Report
Interesting subject, study is well done, valuable in medicine, and medical prctice.
Author Response
Reviewer 2
Dear Reviewer,
Thank you for the revision of our manuscript, and I am very glad that you liked it. The methods were improved in some area. Again, thank you very much for your appreciation.
Best regards,
Agnieszka Kolasa-Wołosiuk
Reviewer 3 Report
The study by Kur et al. highlights hepatic problems in offspring from male rats treated with Finasteride. The authors show an effect in hepatic glycogen content in the offspring. Indeed, the F1Fin rats presented an increased glycogen content, which did not correlate to Glut2 expression. GLUT2, IR and AR were decreased. T and DHT were found to be inversely correlated to androgen receptor. Finally, in adult F1Fin rats hepatic steatosis was detected.
The data presented by the authors is certainly relevant as treatment with this drug seems to have an effect on the offspring. Since this drug is used by patients, it is of great importance to understand and to highlight this transgenerational effects. However, the presented mechanistic part behind this effect is highly speculative. Additional data is needed in order to support the claims.
Many key analysis are missing in the manuscript
- The circulating glucose levels of F1Control and F1Fin?
- Expression of key genes in glycogen synthesis and degradation. Is the upregulation of glycogen content due to lack of degradation or more synthesis of glycogen? This needs to be proven, not extrapolated.
- Is the insulin-dependent AKT pathway signalling changed?
- Are other parameters of hepatic metabolism altered? The authors say that they observe steatosis. Possibly lipid synthesis / degradation that could contribute to the phenotype? This can be assessed by measurement of the hepatic expression of enzymes like Fas, Acc, Scd1, Cpt1, Acox1, but also transporters of lipids in the liver (ex. Cd36). This is a key information, more so, since the authors mention glycogen storage disease in the discussion, which not only is characterized by abnormal glycogen accumulation; but also lipid accumulation due to the excess glucose-6 phosphate.
- A not so recent paper has shown that treatment with Fin leads to hepatic problems in the liver of rats (https://diabetes.diabetesjournals.org/content/64/2/447.long); it induces hepatic steatosis via downregulation of fatty acid oxidation, but also gluconeogenesis alteration. Could this contribute to the phenotype in the offspring as well? The authors mention gluconeogenesis at the end of the discussion, but they do not address it in the data presented.
- As this effect is transgenerational, could there be an epigenetic mechanism involved in the transmission of this effect?
- The table 1 contains the same values as the Figure 1 and 2 in your already published paper ‘’Androgen levels and apoptosis in the testis during postnatal development of finasteride-treated male rat offspring’’, you need to cite the paper, not show the same results again.
- At what age was performed the mRNA expression measurement of Glut2, IR and AT in Figure5? Please clarify in the text.
- There are no representative Glut2 IHC panels for Figure 4. Please provide images representative of your quantification.
- Please restructure the Results section in a more classical display. For example, Figures 1, 2 and 3 can all be presented as one figure with different panels. The results regarding Fig 1, 2 and 3 should all be in one paragraph. A title such as ‘’Hepatic glycogen level in F1Fin rats is increased’’ explaining what the result paragraph is demonstrating could be more useful and clarifying. Figures 4, 5 and 6 can present a second figure, with one paragraph on Glut2 and all under the same title, etc.
Author Response
Reviewer 3
Dear Reviewer,
Thank you for your effort to read and correct my manuscript. I would also like to thank you for your valuable comments. Below you can find my answers for your revision. We hope that now the conclusions are better support by the results.
Best regards,
Agnieszka Kolasa-Wołosiuk
1. The circulating glucose levels of F1Control and F1Fin?
RE:The glucose level in blood (serum) of animals was evaluated and results were added to the text of the publication.
2. Expression of key genes in glycogen synthesis and degradation. Is the upregulation of glycogen content due to lack of degradation or more synthesis of glycogen? This needs to be proven, not extrapolated.
Re: In the text we pointed out that the exact cause of elevated glycogen content in the liver of F1:Fin rats – lack of degradation or increased synthesis - have to be evaluated in the future.
3. Is the insulin-dependent AKT pathway signalling changed?
Re: We can’t answer this question, because it was in majority histological work and not so deeply molecular. In the future we are planning to do more detailed analysis based on DNA microarrays and we are planning to include some elements of the signaling pathway.
4. Are other parameters of hepatic metabolism altered? The authors say that they observe steatosis. Possibly lipid synthesis / degradation that could contribute to the phenotype? This can be assessed by measurement of the hepatic expression of enzymes like Fas, Acc, Scd1, Cpt1, Acox1, but also transporters of lipids in the liver (ex. Cd36). This is a key information, more so, since the authors mention glycogen storage disease in the discussion, which not only is characterized by abnormal glycogen accumulation; but also lipid accumulation due to the excess glucose-6 phosphate.
Re: The current work concerned the carbohydrates metabolism, while the lipid metabolism will be the subject of another publication, because we have already done qRT-PCR for: fatty acid synthase (FASN), acetyl-CoA carboxylase alpha (ACCα), peroxisome proliferator-activated receptor gamma (rPPARG). We are also planning to do IHC for the presence of markers in the liver such as: ACCα, fatty acid synthase, SREBP-1, SCD. We have already purchased these antibodies, now we need time to perform those immunohistochemical reactions, analyze them and compare them with the qRT-PCR results. Therefore, please note that this project is spread over several years and for the publication of various manuscripts; the first stage is the publications on carbohydrate metabolism, the next one will be about lipid metabolism, and in the meantime, the analysis of DNA microarrays.
5.A not so recent paper has shown that treatment with Fin leads to hepatic problems in the liver of rats (https://diabetes.diabetesjournals.org/content/64/2/447.long); it induces hepatic steatosis via downregulation of fatty acid oxidation, but also gluconeogenesis alteration. Could this contribute to the phenotype in the offspring as well? The authors mention gluconeogenesis at the end of the discussion, but they do not address it in the data presented.
Re: The main purpose of this study was to evaluate the effect of finasteride on filial generation. We are planning to evaluate the tissues obtained from the paternal generation. Of course the health condition of the paternal generation can be also connected with the phenotype of the offspring, however it have to be proven, so we will develop another project. Up to now we showed the influence of finasteride in paternal kidney, as an organ that has its development associated with male reproductive organ development.
6. As this effect is transgenerational, could there be an epigenetic mechanism involved in the transmission of this effect?
Re: The purpose of the next project (we are waiting for the approval and will receive additional funding from the grant) is to perform microarrays on DNA because we hope to show the differences between F1:Control and F1:Fin in transcriptomes (expression of genes related to lipid and carbohydrate metabolism, as well).
7. The table 1 contains the same values as the Figure 1 and 2 in your already published paper ‘’Androgen levels and apoptosis in the testis during postnatal development of finasteride-treated male rat offspring’’, you need to cite the paper, not show the same results again.
Re: We found the presentation with previously published results (references are in description of Table 1, [33]) as justified, since in this publication we correlated the levels of T and DHT with GLUT2, IR and AR mRNAs by Spearman’s rank correlation coefficient.
We believe that leaving the table 1. in the publication will not be self-plagiarized.
8. At what age was performed the mRNA expression measurement of Glut2, IR and AT in Figure5? Please clarify in the text.
Re: Figure 5 shows mRNA for GLUT2, IR and AR after analyzing all transcript data results without age grouping, this information is given in the legend to Fig. 5. In the text there is: “Regardless of the age of the animals, the transcripts of GLUT2, IR and AR levels in homogenates of F1:Fin rats’ livers were changed in comparison to F1:Control animals (Fig. 5).”
9. There are no representative Glut2 IHC panels for Figure 4. Please provide images representative of your quantification.
Re: We decided that those photos are not needed, so we only included graphs that nicely and sufficiently show these differences in relation to low, medium and high GLUT2 immunoexpression. Moreover, these data were calculated in a computer program using an appropriate algorithm for this purpose. I hope that you will accept my decision.
10. Please restructure the Results section in a more classical display. For example, Figures 1, 2 and 3 can all be presented as one figure with different panels. The results regarding Fig 1, 2 and 3 should all be in one paragraph. A title such as ‘’Hepatic glycogen level in F1Fin rats is increased’’ explaining what the result paragraph is demonstrating could be more useful and clarifying. Figures 4, 5 and 6 can present a second figure, with one paragraph on Glut2 and all under the same title, etc.
Re: In our opinion, showing Figures 1-6 separately makes them more clear. Moreover, one of the Reviewers asked for Fig. 1. to be larger.
I hope you accept our decision and request of another Reviewer.
Round 2
Reviewer 3 Report
Dear authors,
I am sorry to say that I do not agree with the comments you provided, therefore I cannot recommend your manuscript for publication. Almost all of my comments were not sufficiently answered.
I don't understand, for example, why the authors do not want to provide the immunohistochemistry images and they respond ''We decided that those photos are not needed, so we only included graphs that nicely and sufficiently show these differences in relation to low, medium and high GLUT2 immunoexpression.''
Another example, in regards to the response for my point 4, the authors respond: ''Therefore, please note that this project is spread over several years and for the publication of various manuscripts; the first stage is the publications on carbohydrate metabolism, the next one will be about lipid metabolism, and in the meantime, the analysis of DNA microarrays.''
Metabolism of lipids and carbohydrates is tightly inter-connected and inter-dependent and if you already have data showing that lipid metabolism is altered, these data need to be included in order to have a quality paper, which is not based as much on extrapolation of conclusions.
I hope you understand my point of view and I wish you best of luck with your future studies.